# Tailoring Titanium Sheet Metal Using Laser Metal Deposition to Improve Room Temperature Single-Point Incremental Forming

**DOI:** 10.3390/ma15175985

**Published:** 2022-08-30

**Authors:** Michael McPhillimy, Evgenia Yakushina, Paul Blackwell

**Affiliations:** 1Department of Design, Manufacturing & Engineering Management, University of Strathclyde, Glasgow G1 1XJ, Scotland, UK; 2Advanced Forming Research Centre, Inchinnan PA4 9LJ, Scotland, UK

**Keywords:** Titanium, sheet metal, incremental forming, additive manufacture

## Abstract

Typically, due to their limited formability, elevated temperatures are required in order to achieve complex shapes in titanium alloys. However, there are opportunities for forming such alloys at room temperature using incremental forming processes such as single-point incremental forming (SPIF). SPIF is an innovative metal forming technology which uses a single tool to form sheet parts in place of dedicated dies. SPIFs ability to increase the forming limits of difficult-to-form materials offers an alternative to high temperature processing of titanium. However, sheet thinning during SPIF may encourage the early onset of fracture, compromising in-service performance. An additive step prior to SPIF has been examined to tailor the initial sheet thickness to achieve a homogeneous thickness distribution in the final part. In the present research, laser metal deposition (LMD) was used to locally thicken a commercially pure titanium grade 2 (CP-Ti50A) sheet. Tensile testing was used to examine the mechanical behaviour of the tailored material. In addition, in-situ digital image correlation was used to measure the strain distribution across the surface of the tailored material. The work found that following deposition, isotropic mechanical properties were obtained within the sheet plane in contrast to the anisotropic properties of the as-received material and build height appeared to have little influence on strength. Microstructural analysis showed a change to the material in response to the LMD added thickness, with a heat affected zone (HAZ) at the interface between the added LMD layer and non-transformed substrate material. Grain growth and intragranular misorientation in the added LMD material was observed. SPIF of a LMD tailored preform resulted in improved thickness homogeneity across the formed part, with the downside of early fracture in a high wall angle section of the sheet.

## 1. Introduction

Single-point incremental forming (SPIF) is a method of producing sheet metal parts without the need for expensive tooling. In place of the conventional fixed forming die setup, a computer numerical control (CNC) machine controls a tool as it performs incremental localised deformation on the sheet [1]. It is thanks to the incremental form of deformation that SPIF is capable of increasing the forming limits of difficult to form materials such as titanium [2], with Jackson et al. [3] highlighting plain strain stretching and Allwood et al. [1] through thickness shear as key deformation modes. Lightweight materials such as titanium are widely used in aerospace and automotive industries for their high strength to weight ratio [4]. However, their alpha-grained hexagonally close-packed (hcp) microstructure limits their formability at room temperature due to a restricted number of slip systems. As such, there are increasing demands for advanced forming technologies to improve the low temperature formability of this lightweight material [5]. The increased forming limits achievable with SPIF makes room temperature forming of titanium possible, offering a cost-effective alternative to forming titanium at elevated temperatures [6].

A disadvantage of SPIF is the localised sheet thinning during SPIF which limits the potential geometric capabilities of the process [7]. A combination of the compressive load induced by the tool and stretching of the sheet was found to be responsible for the thinning [8]. Jeswiet et al. [9] attributed high forming angles to increased levels of thinning in the sheet cross-section which can result in inhomogeneous thickness distributions across SPIF parts. This is especially true in sheet parts containing wall angles at 70° to the sheet plane or higher [10]. Reducing the severity of sheet thinning in room temperature SPIF is integral to the successful forming of titanium sheet parts. Different approaches from other researchers have been taken to achieve this. Mohsen et al. [11] combined SPIF of AA6082-T6 sheet with hydraulic bulging, increasing the formability by 26% and reducing sheet thinning by 45%. Ambrogio et al. [12] machined pockets into specific zones of a AA1050-O aluminium sheet to weaken these areas to better control the thinning distribution during ISF. In another study, Ambrogio et al. [13] used additive manufacturing (AM) to avoid excessive sheet thinning and improve accuracy via two routes, an AM generated backing plate and sintered material to reinforce the area of pronounced deformation in aluminium sheet. Kumar et al. [14] found decreasing the wall angle, step depth and spindle speed and increasing tool diameter reduced thinning, so improving uniform thickness distribution of SPIF parts.

Tailored blanks offer a potential route for generating preforms for optimising forming. Raut et al. [15] designed aluminium and steel blanks with areas of thicker and stronger material in zones of expected higher stress and loading to reduce part weight without compromising strength. AM reinforcements have been seen to offer better final part properties than other types of tailored blanks due to the full metallurgical bonding between deposit and substrate [16]. Laser metal deposition (LMD) is particularly suited for tailoring sheet metal due to its good heat dissipation, its capability of printing thin layers and fast deposition speeds which limit sheet deformation [17]. Such advantages led researchers to propose the use of LMD to increase the thickness and stiffness of aluminium sheet with locally applied patches [16]. The material properties of LMD parts can be advantageous to specific applications as found in the following studies. Yu et al. [18] discovered LMD generated Ti6Al4V parts resulted in superior yield and ultimate tensile strengths than cast and annealed wrought material, with consistent microstructure between layers and tracks and a low dilution depth and heat-affected zone (HAZ). Barro et al. [19] found AM generated titanium grade 4 parts for dental components performed better than milled parts. LMD is capable of processing a wide range of metals making it the most versatile AM process [20]. With LMD the geometry is printed on a substrate surface by melting the surface with a laser whilst simultaneously applying metal powder. The melt pool is protected from oxidation by an inert gas, usually argon or helium, and the metal powder is fed into the melt pool by a nozzle. Typically, the part is kept stationary as the deposition head follows a tool path to achieve the final part geometry.

The purpose of this investigation is to tailor a commercially pure titanium (CP-Ti) SPIF preform with local thickening via LMD to improve thickness homogeneity in the final SPIF sheet part. The development of this novel SPIF + LMD hybrid process may potentially provide a route to improve geometric accuracy at high forming angles in cold formed CP-Ti sheet parts. A commercially pure titanium grade 2 (CP-Ti50A) sheet will be locally thickened with CP-Ti powder using LMD. The two variables of interest are LMD build height (in the build direction) and LMD track direction (with respect to the sheet rolling direction, RD). The impact which LMD build height and direction have on the tailored materials mechanical and directional properties is to be examined. Tensile testing will be performed to examine the mechanical behaviour of the tailored material with in-situ digital image correlation (DIC) to measure surface strain distribution. Material characterisation techniques will be used to examine the microstructure of the tailored material. Following this, a forming trial will be performed to test the capability of the LMD tailored preform for room temperature SPIF. LMD variables are to be kept constant to avoid confounding factors from influencing the results of the study. All the details are reported in the following sections.

## 2. Materials and Methods

The sheet used for the LMD substrate material was an uncoated cold rolled sheet of CP-Ti50A of dimensions 500 mm × 500 mm × 1.6 mm (W × L × T), selected for its good room temperature ductility relative to other titanium alloys. The chemical composition of the material is listed in Table 1 [21] and its mechanical properties in Table 2. Examination of the as-received (A-R) sheet substrate showed a microstructure of fine equiaxed grains with average grain size of 5.4 ± 0.2 μm, Figure 1a. The true stress–strain curve of the material in three directions with respect to the sheet rolling direction (RD) showed anisotropy of the material, Figure 1b. The powder metal used for laser metal deposition (LMD) was a CP-Ti grade 2 powder with particle size distribution of 45–150 μm. It was gas atomised with a spherical particle shape, apparent density 2.62 $g/{\mathrm{cm}}^{3}$ and oxygen content 0.16% [22].

A non-contact blue light 3D scanning technique was used to scan a 1.6 mm thick CP-Ti50A sheet part previously formed by SPIF (SPIF part). The part was designed with wall angles (θ) at 20°, 40°, 60° and 80° to the sheet plane. GOM ATOS^®^ software was used to generate a three-dimensional representation of a section of the part, Figure 2. Through-thickness thinning was measured to an accuracy of ±20–40 μm across each angled section. Added thickness from a spray, used to minimise light reflection, was responsible for an additional 0.05 mm thickness. Five measurements were taken from each section and the mean calculated, Table 3. The thinning measurements determined the quantity of material to be added by LMD. The same scanning technique was used to generate a digital model of the SPIF + LMD hybrid sheet part created as part of this study, to measure its thickness profile.

Two LMD tailored sheets were designed for this investigation. The first for conducting material analysis on the LMD tailored material, Figure 3, and the second a LMD tailored preform for SPIF, Figure 4c. Designing the first sheet with nine LMD deposit (pad) locations made it possible to investigate the influence of the LMD build height and orientation on the materials mechanical and microstructural properties. Each pad measured 111 mm × 71 mm (L × W). Three pads were orientated with their length parallel to the sheet rolling direction (RD), three at 45° to the RD (45° to RD) and three at 90°, or transverse, to the RD (TD). From each pad three tensile specimens were extracted. The added thickness via LMD (A) in the build direction (Z) was achieved by stacking 0.15 mm layers in a crosshatch pattern (ex. x4 layers achieves 0.6 mm added thickness). The tensile specimen gauge lengths were made in-line with the LMD pad length (Y) which determined the tensile specimen orientation in relation to the sheet rolling direction (RD). The resulting material variable configurations are provided in Table 4. Additional tensile specimens were designed to be extracted directly from the A-R sheet for testing (sets a, e and i). For the SPIF preform a uniform thickness of 0.3 mm was deposited using the same LMD parameter across a second CP-Ti50A sheet. The deposited material aligned with the areas of thinning in the previously formed SPIF geometry.

A modular fixture consisting of two sub-modules was designed to support the hybrid LMD + SPIF process. One sub-module (LMD fixture) secured the sheets during LMD and a subsequent stress relief heat treatment. The LMD fixture comprised of three parallel plates made from hardened tool steel with a central cavity of 442 mm diameter in the top plate to allow the die to pass through during SPIF. The LMD fixture, with sheet clamped, fits into the larger sub-module (SPIF fixture). The SPIF fixture positions and lowers the LMD fixture over a die as the sheet is formed with a 25.4 mm diameter hemispherical tool. Physical alignment grooves and guiding pillar/bushing assemblies in the SPIF fixture ensure accuracy and repeatability.

LMD was performed by a commercial supplier using a TRUMPF 2 kW TruDisk Laser and Oerlikon Metco Twin-10-C powder feeder. The laser and powder feed movement were provided with a 5-axis REIS RL80 Gantry Manipulation System with a 3 m × 2.25 m × 1.5 m work envelope, Figure 4a. The following LMD machine parameters were developed experimentally on-site: laser power (P) of 675.0 W, scanning speed (v) of 15.0 mm/s, track separation of 1.4 mm, and carrier and nozzle shielding gas flow rates of 4.0 L/min and 10.0 L/min, respectively. These parameters achieved a low linear laser energy density (LED) of 45 J/mm ensuring a low depth of heat penetration, a narrow melt region and minimal HAZ [23]. The shielding (argon) gas carried the powder into the molten pool where it was melted and, on solidification, layered with a step size of 0.15 mm. These machine parameters were used for performing LMD on both sheets, Figure 4b,c.

X-ray diffraction (XRD) was used to measure the residual stress remaining after LMD. The results of XRD showed some variation in the stress measurements acting in the RD and TD directions, Table 5. The maximum stress was made to be 168 ± 6.69 MPa, below the typical yield strength (YS) of CP-Ti50A which is 345 MPa [21].

A stress relieving heat treatment was performed using a Carbolite LCF furnace to ensure no sheet movement affected the LMD build quality. Following this, tensile specimens were extracted by wire electrical discharge machining (EDM) from the LMD part using a AgieCharmilles CUT 400 Sp machine. The geometry of the tensile specimens is provided in Figure 5. A speckle pattern was spray painted on the surface of each tensile specimen for digital image correlation (DIC) analysis, used to track the sample surface displacement and generate an accurate strain map. Selected specimens had the speckle pattern applied to the LMD deposit side and others on the substrate side.

A 150 kN Zwick/Roell Z150 testing machine was used to perform room temperature (23 °C) uniaxial tensile tests. All tests were carried out at a constant strain rate of $0.001{\;s}^{-1}$, controlled by testXpert II software and in accordance with ISO 6892-1:2019. An extensometer measured the elongation of the specimen. The test ended at tensile specimen failure. Twelve sets of tensile specimens were extracted from the LMD tailored sheet with different configurations of the LMD build height and build orientation, see Table 4. For each configuration, three repeat tests were performed to validate results. In total, 36 tensile tests were performed. In-situ DIC analysis measured local surface strain during tensile deformation. A two-camera setup captured strain data with a fixed frequency of 1 Hz and strain maps were calculated by measuring the displacement of the surface speckle pattern using DaVis 8.0 software supplied by LaVision.

Following mechanical testing, material characterisation of the LMD tailored material was conducted. The LMD surface morphology was analysed using an Alicona Infinite Focus microscope. The tensile samples were mounted, cleaned with acetone and the fracture face examined using a Quanta 250 FEG SEM. Following this, grinding and polishing steps were performed to prepare the sample surfaces for further characterisation. Material hardness was measured using a Vickers Micro Hardness Tester by applying a force of 1 kg for 15 s using a diamond indenter. Samples were polished and the surface etched to expose the grain boundaries for optical microscopy. An electropolishing step was used to prepare material samples for microstructural examination. A polishing current of 0.14 A and voltage 35 V was used to polish a 1 cm^2^ area with A3 electrolyte. Electron backscatter diffraction (EBSD) was performed with step size 0.2 μm to observe the material microstructure using an SEM operated with accelerating voltage of 20 kV.

Room temperature SPIF was performed on the LMD + SPIF preform, Figure 6. A DMU 125 FD Duo Block 5-axis CNC machine was used to perform SPIF. The SPIF fixture was secured to the machine bed, Figure 6a. Scissor lifts lifted the top plate of the SPIF fixture into position above the die, the LMD fixture with the LMD + SPIF preform attached was installed into the SPIF fixture, and the scissor lifts removed. The toolpath was generated using Autodesk Fusion 360. Vericut was used to test for collisions and the programme uploaded to CIMCO and made available to the CNC machine operator. SPIF was stopped after every 10 mm of forming and the CNC machine door unlocked to check the part for signs of failure, measure the temperature of the tool, and lubricate the sheet and tool. SPIF was completed and the formed part (LMD + SPIF part) taken for analysis, Figure 6b.

## 3. Results

### 3.1. Microstructural Response to LMD

Optical microscopy was used to view the microstructural development across the cross-section of the A-R, Figure 7a, and LMD material samples, Figure 7b,c. Near-surface grain growth is observable in the LMD tailored samples to a depth equal to the added thickness (Zone 1), Figure 8a,d. Near the base of each sample there is a fine equiaxed microstructure which is indicative of the non-transformed substrate material (Zone 3), Figure 8c,f. The microstructure in Zone 3 is comparable to the A-R material, Figure 7a. At the interface between Zone 1 and 3 is the HAZ (Zone 2), Figure 8b,e. The microstructural change in the HAZ was likely a result of the thermal input from the laser during LMD. With the equipment available it was not possible to measure the temperature during deposition, however it is expected to have risen above the CP-Ti transus temperature of 882 °C [24]. This thermal input from the laser and subsequent cooling is likely the cause for the α > β > α phase change. As such, Zone 2 contains a platelet alpha microstructure, typical for CP-Ti when cooled from the high temperature body centered cubic (bcc) β-phase field [25]. An increase in the HAZ thickness from 0.9 mm to 1 mm and a decrease in the un-transformed substrate from 0.7 mm to 0.5 mm is observed in response to the additional LMD added thickness, Figure 7b,c. 

The difficult sample preparation process for CP-Ti meant the microstructure was not clearly visible by optical microscopy. As such, EBSD analysis was performed on samples extracted from the LMD thickness to view its microstructure in greater detail. Grinding to depths of 0.3–0.6 mm below the material surface exposed the microstructure within Zone 1. Inverse pole figure (IPF) maps were generated, Figure 9. The RGB colour code: red for (0001), green for (12$\bar{1}$0) and blue for (01$\bar{1}$0) as shown in the standard stereographic triangle corresponds to the crystallographic orientation of each grain. The microstructure of the LMD part remains randomly orientated, Figure 9b,c, similar to the A-R microstructure, Figure 9a. Low magnification maps were used to measure the average grain size of the LMD tailored material using the intercept method. For the 0.3 mm thickened material, 158 grains were measured with an average grain size of 54.5 ± 0.2 μm and size range of 8.1 μm–250 μm (SD = 56.9). For the 0.9 mm thickened material 158 grains were measured with an average grain size of 39 ± 0.2 μm and size range of 2.2 μm–326.2 μm (SD = 57.7). The A-R material has an average grain size of 5.4 ± 0.2 μm (SD = 3.3). A grain growth of approximately x10 is observed in response to LMD. The higher SD of the LMD tailored microstructure suggests significant spread in grain size, in contrast to the lower SD of the A-R material which suggests a normal distribution of grain size. Twinning was observed in the LMD material. Some twins are seen to have nucleated and terminated on grain boundaries, labelled T1 in Figure 9d, whereas other twins nucleated on a boundary and terminated within the grain interior, labelled T2. Very small amounts of leftover transformed beta (β) phase is present in the LMD tailored material, likely a leftover during a phase transition in response to the LMD heating cycle.

A high magnification coloured IPF map was performed to examine the microstructure of the LMD part in greater detail, Figure 10a. The map contains several single grains and a fragmented large grain with intragranular misorientations. The single grains are presumed to be the microstructure of the melted deposit material. The fragmented grain has an internal substructure containing partially and fully enclosed grains with low angle grain boundaries (LAGB) represented by blue lines, and several fully enclosed grains with high angle grain boundaries (HAGB) with misorientation over 15°, represented by black lines. The substructure has a single crystal orientation as determined by the IPF colouring. A misorientation angle measurement across a fully enclosed HAB grain, labelled L1 in Figure 10b, reveals that the misorientation is just over 15° which confirms the internal HAGBs evolved from LAGBs with the accumulation of dislocations. An incomplete LAGB misorientation, labelled L2 in Figure 10c, has a misorientation of just under 8° which suggests further accumulation of dislocations could result in a fully enclosed LAB grain which could develop into a HAB grain. The size and shape of the LAB segments are close to the fine equiaxed microstructure of the A-R CP-Ti50A material, Figure 9a. The occurrence of incomplete grains is a strong indication that continuous dynamic crystallisation (CDRX) occurred [26].

### 3.2. Mechanical Response to LMD Tailoring

Tensile test specimens from the tailored sheet were prepared along the RD, 45° to RD and TD directions for uniaxial tensile testing. The tensile test for each specimen was repeated three times and averaged. The tensile test data was plotted as shown in Figure 11a,c,e and the tensile specimens at fracture are seen in Figure 11b,d,f. The details of the tensile test are tabulated in Table 6. The CP-Ti50A sheet material in the A-R form exhibits anisotropic tensile behaviour, Figure 1b. In the RD, the A-R material has a higher ultimate tensile strength (UTS) and greater elongation before necking and fracture than at 45° to RD. This anisotropy is a direct response to the pronounced texture and limited number of slip systems in hexagonally close-packed (hcp) materials such as CP titanium. Despite the anisotropic tensile behaviour of the substrate material, the specimens extracted from the LMD part exhibited isotropic mechanical behaviour. This result suggests LMD tailoring of sheet preforms will allow for good design flexibility due to the omnidirectional mechanical behaviour across the thickened preform. In the RD, the LMD thickened material specimens showedreduced strength and a small reduction in ductility with more abrupt hardening suggesting increased brittleness compared to the A-R material, Figure 11a. The results show a 2.4–5.6% increase in the yield stress (YS), 4–6% reduction in elongation (E) and a 6.2–9.9% decrease in the ultimate tensile strength (UTS) compared to the A-R material. At 45° to the RD the LMD material exhibited a 5.2–9.9% increase in the YS, 11–14% reduction in E which indicates lower ductility, and similar UTS values of a range 0–4.6% higher than the A-R material, Figure 11c. In the TD there was little change in the mechanical behaviour of the LMD tailored material compared to the A-R material with a 1.1–2.7% decrease in the YP, 3–6% decrease in the E and a 3.5–5.3% decrease in UTS values, Figure 11e. In general, the results indicate that LMD thickening reduces elongation and formability. A larger decrease in ductility was seen in the RD and 45° to RD directions. A noteworthy observation is the lack of influence the number of stacked LMD layers had on the strength properties of the thickened material. This suggests the major change in the crystallographic microstructure occurred at the initial introduction of the LMD material and remained constant as more layers were deposited. The tensile results indicate there are unlikely to be significant defects in the added LMD material, such as pores. The material performs well and would suggest the introduction of an added thickness via LMD is a potential route for tailoring preforms for SPIF. In summary, the isotropic nature of the LMD tailored material and lack of influence of the LMD build height ensure that anisotropy and geometry are less relevant when designing LMD tailored SPIF preforms.

### 3.3. Strain Analysis of LMD Material

Optical non-contact measurement techniques were used to show effective strain displacement in the plastic region of the tensile specimens during tensile deformation. At 90% extension in the A-R RD sample, strain was seen to concentrate near the gauge centre and localised necking was observed, followed by diffused necking and reduction in the gauge width before failure. High strain localisation was seen to initiate at the edge of each sample before failure, at which a crack nucleated and travelled through the specimen. The DIC results indicate the necking morphology changed in response to the introduction of an LMD layer. With the specimens from the LMD part, strain localised in a smaller area during necking and eventual fracture. The A-R specimen had a diffused neck of 10 mm and localised neck of 5 mm at 90% extension, Figure 12a. The LMD tailored specimens in the RD had diffused necks of 5 mm and localised necks of 1–1.5 mm at approximately 90% extension, Figure 12b–d. Strain was seen to localise within shallow grooves between LMD tracks compared to the homogeneous strain distribution across the surface of the AR specimen. Similar strain behaviour was observed for all samples oriented 45° to RD and TD, Figure 13 and Figure 14. This suggests the surface morphology from the LMD tracks influenced strain distribution. The LMD tailored specimens exhibited lower elongation and earlier fracture nucleation and failure compared to the A-R specimens as indicated by the true stress–strain curves. This is likely a result of cracks nucleating from defects in the heterogeneous LMD surface, causing necking to initiate at earlier strain during tensile deformation in comparison to the more homogeneous A-R material specimens.

A speckle pattern was sprayed on the reverse side of selected specimens to compare the strain distribution on the LMD surface (front) and reverse side with no LMD (back), Figure 15. At 90% extension the diffused neck measured 5 mm on both front and back of the RD and TD specimens with 0.9 mm added LMD material. Despite the same diffused necking, the high strain which indicates localised necking was 1–2 mm on the front and 3–4 mm on the back for the RD and TD specimens. This indicates surface morphology likely impacted surface strain distribution. Higher surface strain at 90% extension occurred on the front compared to the back, with 0.80 and 0.90 strain values for front RD and TD specimens, respectively, compared to 0.72 and 0.68 strain values for back RD and TD specimens, respectively.

### 3.4. Material Study

Due to the influence of the LMD deposit morphology on the surface strain distribution, an Alicona microscope was used to measure the depth of grooves between each LMD track. A 3D representation of the surface was stitched together from layered images using OmniSurf 3D software, Figure 16a. The LMD surface morphology of the analysed areas contain parallel LMD tracks with relatively flat surfaces and shallow grooves. The average groove depth across all LMD sample surfaces was 73.5 ± 10.6 µm. Unconsolidated CP-Ti powder was seen to collect in the track valleys, Figure 16b. This is likely a result of this region being located at the cooler outer regions of the laser melt pool. The unconsolidated powder particles may have functioned as surface defects, acting as crack nucleation sites for early onset fracture. Machining is a potential route to improve the quality of the final LMD surface.

Fractography analysis was performed on the fracture face of the A-R tensile specimen and the sample with 0.9 mm added thickness by LMD, both with the gauge length orientated in the sheet RD. Fractography was performed using the FEI Quanta 250 FEG SEM at high magnification. The centre of the fracture face on the A-R sample (Zone 1) is characterised by a fine dimple texture accompanied by microscopic voids typical of ductile fracture, Figure 17b. The edge of the A-R sample has small dimples, micro-voids and evidence of shear, Figure 17c. The LMD surface (Zone 1) and non-transformed substrate material (Zone 2) have distinct fracture morphologies, Figure 18a. No delamination of the LMD material is evident with good bonding across the interface region of Zone 1 and Zone 2. Within Zone 1 are large equiaxed voids with groups of coalesced microscopic voids at their base, Figure 18b. Zone 2, Figure 18c, has similar fracture morphology to the edge of the A-R specimen, Figure 17c, with micro-voids propagating in the direction of the stress axis and visible surface shear, suggesting the microstructure in Zone 2 was not altered significantly by LMD.

Vickers hardness testing of the A-R material and LMD part was performed. For the in-plane measurements a depth of 280 μm was removed by grinding to measure the microhardness of the added LMD layer, Figure 19a. The hardness results for the A-R material are highly repeatable with a mean of 153 ± 3 HV. The hardness results for the tailored materials exhibit large scattering which is likely due to the inhomogeneous microstructure caused by the grain growth [18]. The mean hardness values for the tailored materials are in the range of 159–166 HV with significant overlap of the standard deviation error bars showing the difference is not statistically significant. This indicates the hardness is consistent across the tailored material despite the difference in LMD build thickness. Microhardness measurements were performed along the gauge length of the fractured tensile samples, from the top grip region towards the fracture site, Figure 19b. A rise in microhardness across the gauge of approximately 173–182 HV was observed across all material samples.

### 3.5. SPIF of LMD Tailored Preform

SPIF was performed on the LMD + SPIF preform until fracture. A three-dimensional representation was generated for the LMD + SPIF preform, Figure 20a, and the LMD + SPIF part, Figure 20b. The colour map shows thickness variation.

The thickness distribution was calculated across the SPIF part and LMD + SPIF parts, Figure 21a. The rise in thickness at the edge of the unformed section for the LMD + SPIF part is the unformed LMD thickened material. The results indicate the percentage (%) rate of thinning is equal for both formed sheets when measured directly from the as-received starting thickness, Figure 21b. However, when measured from the starting thickness of the reinforced wall sections, the % rate of thinning is lower for the LMD + SPIF part compared to the SPIF part, Figure 20c, suggesting greater thickness homogeneity in SPIF parts is achievable by using tailored preforms with additional thickness in regions of expected thinning.

Despite the results indicating improved thickness homogeneity, the LMD + SPIF preform exhibited cracks in the LMD surface and eventual fracture in the 60° wall angle section. In comparison, the SPIF part without LMD tailoring failed in the 80° wall angle section. Further analysis of the LMD + SPIF part is planned to better understand the cause of this failure with the goal of optimising the hybrid SPIF + LMD process.

## 4. Summary and Conclusions

The aim of this work was to generate a LMD tailored preform with variable thickness to mitigate thinning, a common defect in room temperature SPIF of titanium parts with high angled walls. An initial material study of a LMD tailored CP-Ti50A sheet with localised thickening was performed. Following this, SPIF was performed on a LMD tailored CP-Ti50A preform sheet. To facilitate the hybrid LMD + SPIF process a modular fixture was designed to constrain the tailored titanium sheet during LMD, post-processing, and SPIF.

The main findings of this work are summarised below:Microstructural analysis of the LMD tailored material showed distinct regions across the thickness of the material sample. This consisted of the LMD deposited layer, a HAZ interface zone and the non-transformed substrate material.EBSD analysis of the LMD tailored material showed grain growth in response to LMD. Fragmentation of large grains was observed with intragranular LAB misorientations and fully enclosed HAB grains.Isotropic mechanical properties were observed in the LMD tailored material in contrast to the anisotropic properties of the A-R CP-Ti50A sheet. The LMD build height appeared to have little influence on strength. As such, an SPIF preform can be designed without significant consideration of directionality, or concerns about the thickness of LMD material. The results indicate LMD reinforcement reduces material ductility and lowers formability of CP titanium.In-situ DIC analysis during tensile testing showed the LMD build geometry to alter effective strain distribution across the sample surface, with surface strain propagating parallel to the LMD beads. This differs from the homogeneous spread of surface strain across the A-R samples.Fractography analysis of the LMD thickened material fracture face showed a population of dimples and fine microscopic cracks, accompanied by large equiaxed voids with internal coalesced microscopic voids. No delamination of the deposit layer was evident suggesting good cohesion between the LMD layer and substrate.Microhardness values measured across the tailored material were statistically equal despite the difference in LMD build thickness.LMD tailoring was seen to achieve greater homogeneity in the SPIF part, however the part failed during forming the 60° wall angled section.

The result of this study indicates tailoring preforms with variable thickness offers a potential route to optimise room temperature SPIF of titanium parts. However, issues regarding this hybrid process have been discovered which must be addressed. One such issue is the limited formability of the LMD tailored material which limits the achievable wall angle for the SPIF part. A subsequent quality assessment and analysis of the fracture modes of the final sheet part is planned.

## Figures and Tables

**Figure 1 materials-15-05985-f001:**
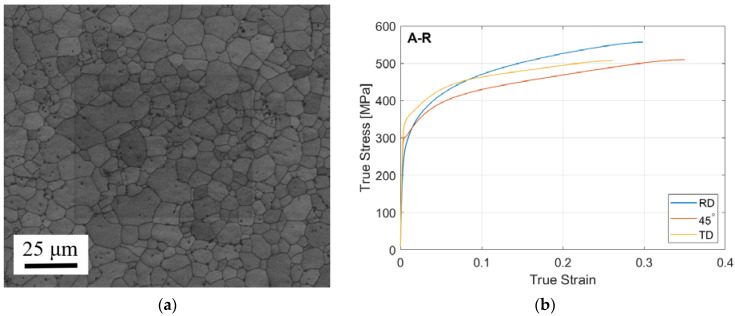
Substrate material properties: (**a**) Microstructure of A-R (CP-Ti50A); (**b**) True stress–strain plot of A-R CP-Ti50A in three directions at room temperature.

**Figure 2 materials-15-05985-f002:**
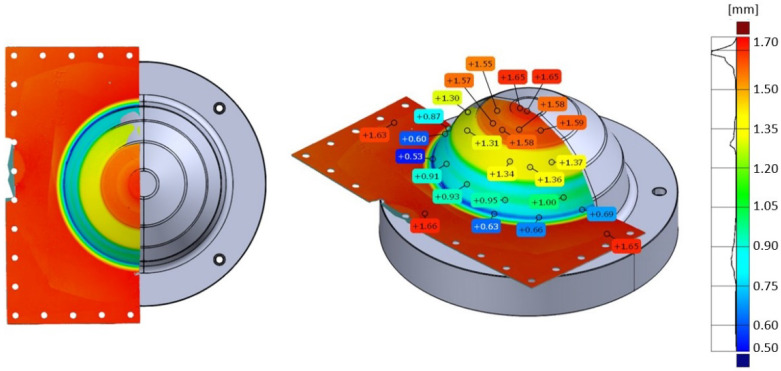
GOM scan to measure thickness reduction across SPIF formed part.

**Figure 3 materials-15-05985-f003:**
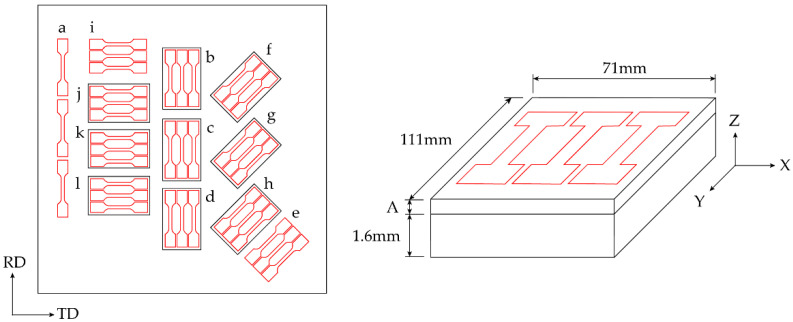
LMD pad locations (internal black boxes) on sheet with the tensile test specimen extraction locations (red) indicated, and tensile gauge lengths in-line with pad length (Y).

**Figure 4 materials-15-05985-f004:**
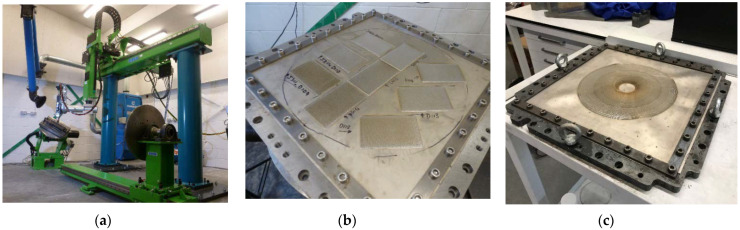
LMD fixture and sheet for LMD: (**a**) LMD processing cell. (**b**) Fixture with CP-Ti50A sheet mounted on T-slot table; (**c**) Completed LMD deposits on sheet.

**Figure 5 materials-15-05985-f005:**
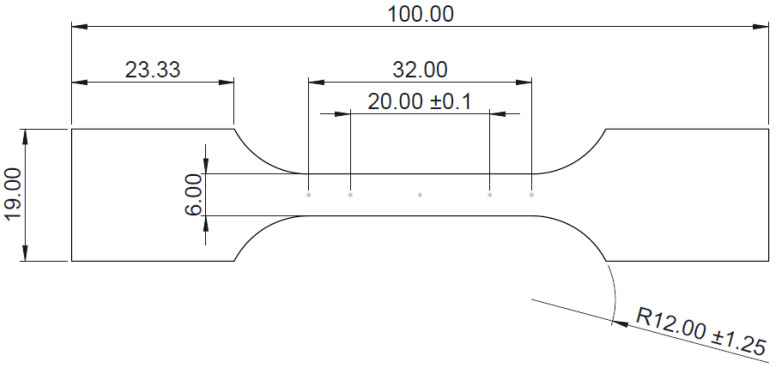
Geometry of tensile specimen based on ASTM-E8/EM (dimensions in mm).

**Figure 6 materials-15-05985-f006:**
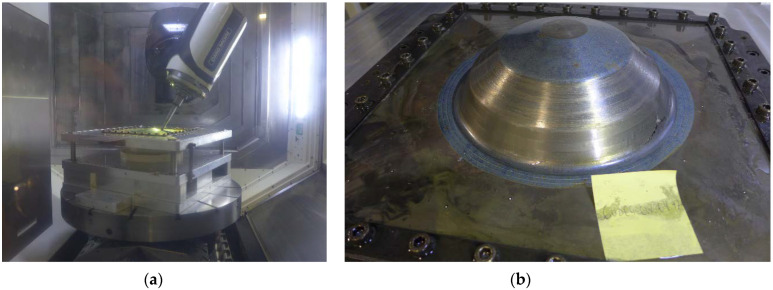
Performing SPIF on LMD tailored part: (**a**) Modular fixture with sheet attached during SPIF. (**b**) Final SPIF part with fracture.

**Figure 7 materials-15-05985-f007:**
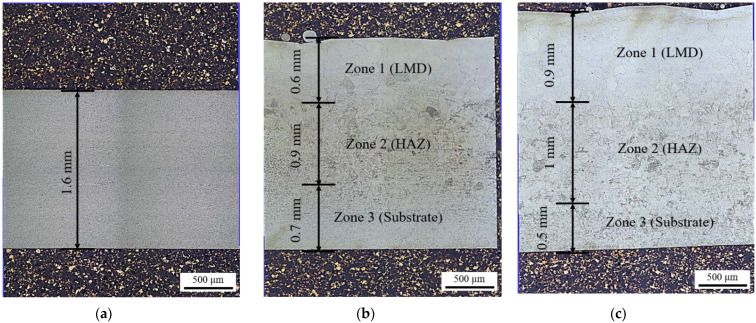
Optical images of material sample cross-sections at ×10 magnification with approximate LMD layer, heat affected zone (HAZ) and unchanged substrate defined: (**a**) A-R material; (**b**) 0.6 mm LMD sample; (**c**) 0.9 mm LMD sample.

**Figure 8 materials-15-05985-f008:**
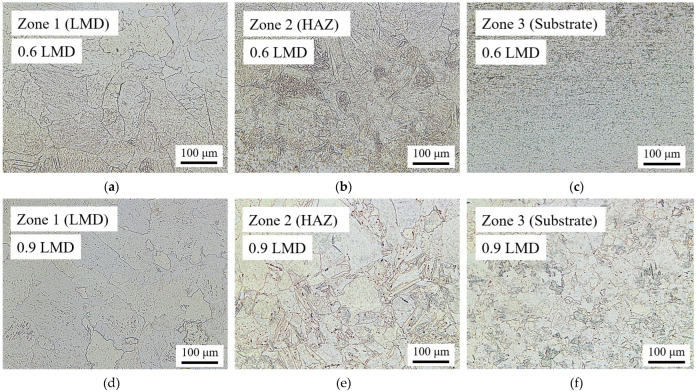
Optical images of 0.6 mm and 0.9 mm LMD samples across cross-section at ×20 magnification: (**a**) Zone 1 (LMD tailored material) of 0.6 mm LMD sample; (**b**) Zone 2 (Heat affected zone, HAZ) of 0.6 mm LMD sample; (**c**) Zone 3 (Non-transformed substrate) of 0.6 mm LMD sample; (**d**) Zone 1 (LMD tailored material) of 0.9 mm LMD sample; (**e**) Zone 2 (Heat affected zone, HAZ) of 0.9 mm LMD sample; (**f**) Zone 3 (Non-transformed substrate) of 0.9 mm LMD sample.

**Figure 9 materials-15-05985-f009:**
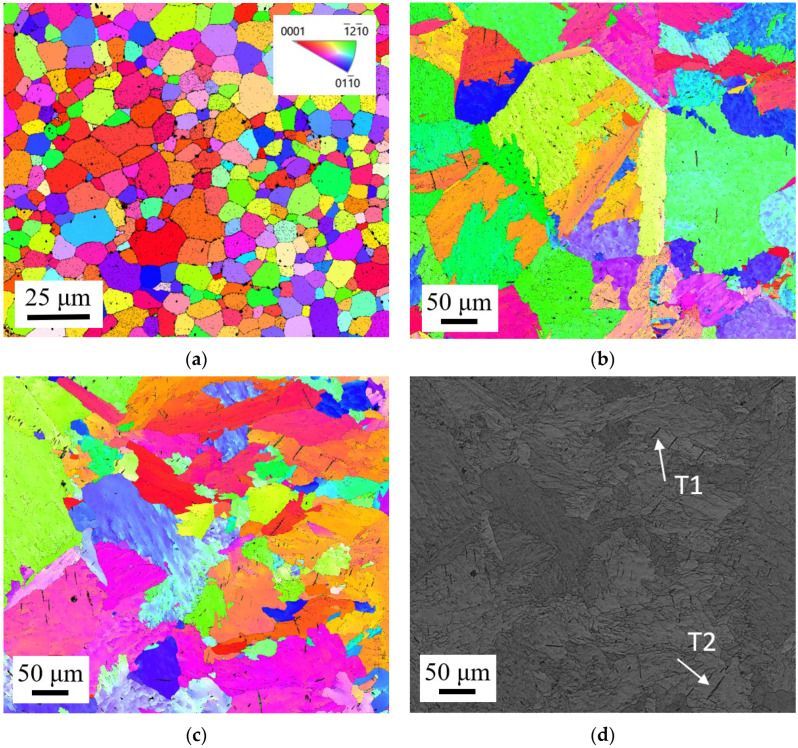
EBSD analysis of material specimens extracted from LMD tailored material: (**a**) IPF map for A-R specimen with no LMD thickening (**b**) IPF map of 0.3 mm LMD layer; (**c**) IPF map of 0.9 mm LMD layer; (**d**) Band contrast image of 0.9 mm LMD layer showing twinning.

**Figure 10 materials-15-05985-f010:**
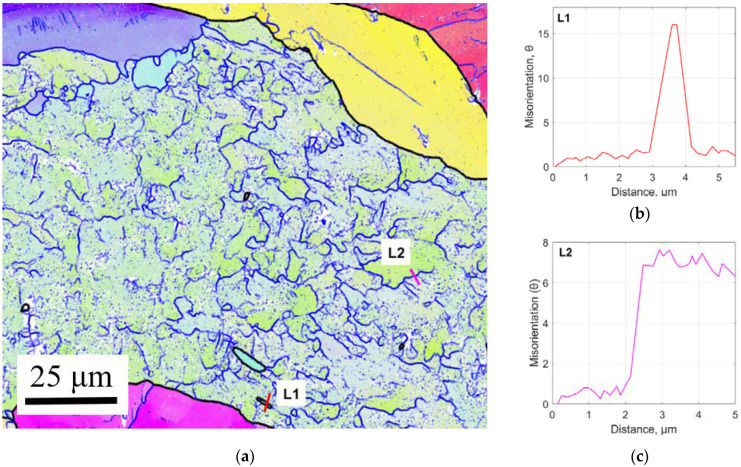
EBSD analysis of LMD tailored metal: (**a**) x2000 magnified coloured IPF map of 0.9 mm LMD layer; (**b**) misorientation across fully enclosed HAGB; (**c**) misorientation across segmented LAGB.

**Figure 11 materials-15-05985-f011:**
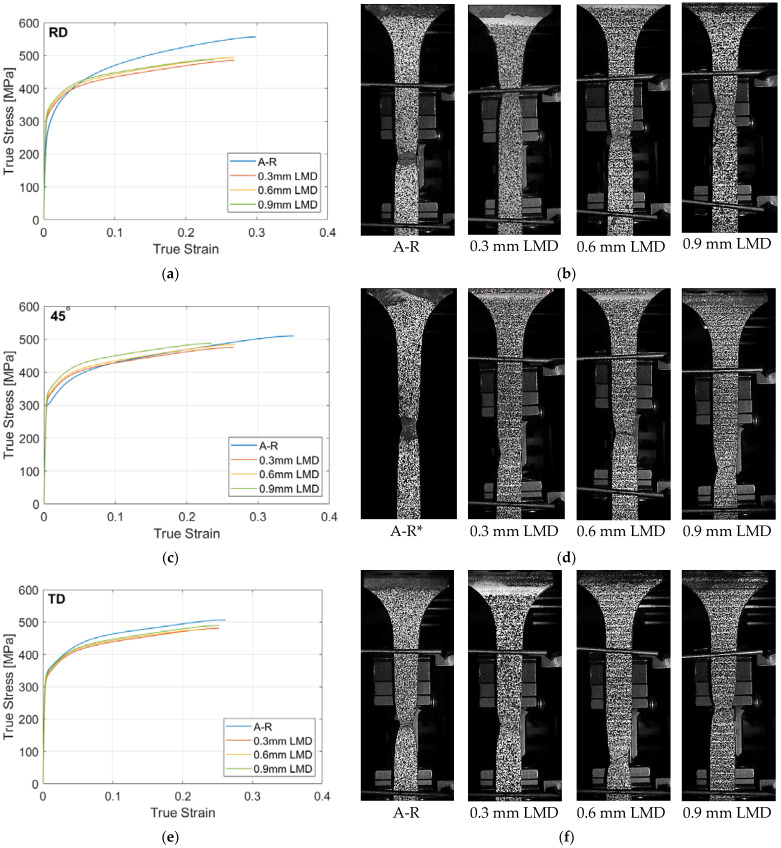
Results of uniaxial tensile testing: (**a**) True stress (α) vs. strain (ε) curves in the sheet RD; (**b**) RD tensile specimens at fracture; (**c**) True α vs. ε curves in the diagonal direction (45°); (**d**) 45° diagonal tensile specimens at fracture * Extensometer was removed due to amount of elongation and plot was calculated from raw strain data; (**f**) True α vs. ε curves in the TD direction; (**e**) TD tensile specimens at fracture.

**Figure 12 materials-15-05985-f012:**
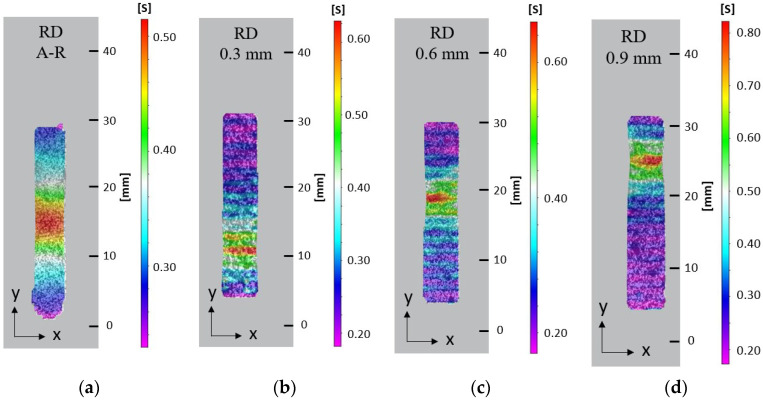
Effective surface strain at 90% extension on tensile samples orientated in substrate RD: (**a**) A-R material; (**b**) 0.3 mm LMD sample; (**c**) 0.6 mm LMD sample; (**d**) 0.9 mm LMD sample.

**Figure 13 materials-15-05985-f013:**
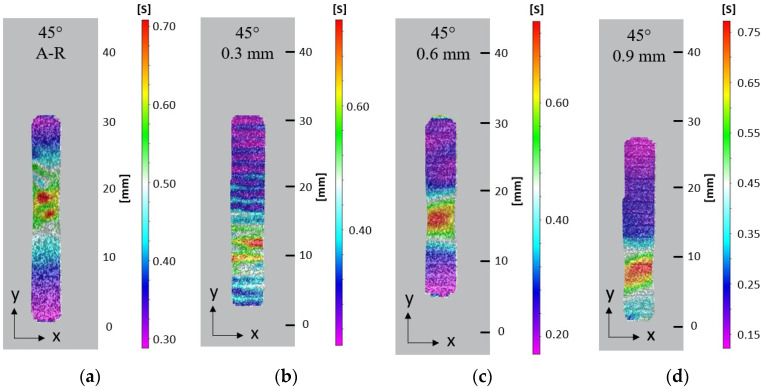
Effective surface strain at 90% extension on tensile samples orientated 45° to substrate RD: (**a**) A-R material; (**b**) 0.3 mm LMD sample; (**c**) 0.6 mm LMD sample; (**d**) 0.9 mm LMD sample.

**Figure 14 materials-15-05985-f014:**
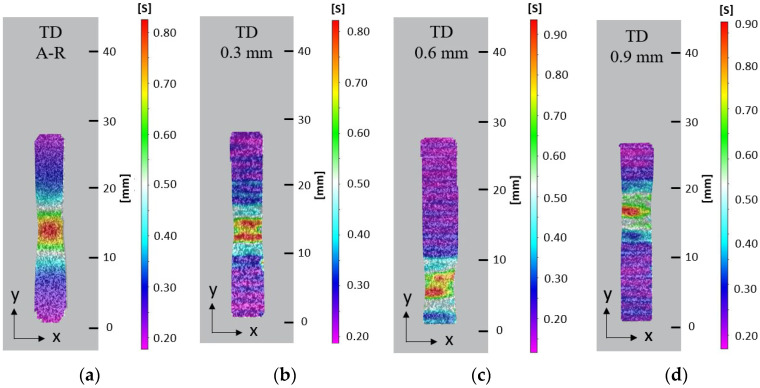
Effective surface strain at 90% extension on tensile samples orientated in substrate TD: (**a**) A-R material; (**b**) 0.3 mm LMD sample; (**c**) 0.6 mm LMD sample; (**d**) 0.9 mm LMD sample.

**Figure 15 materials-15-05985-f015:**
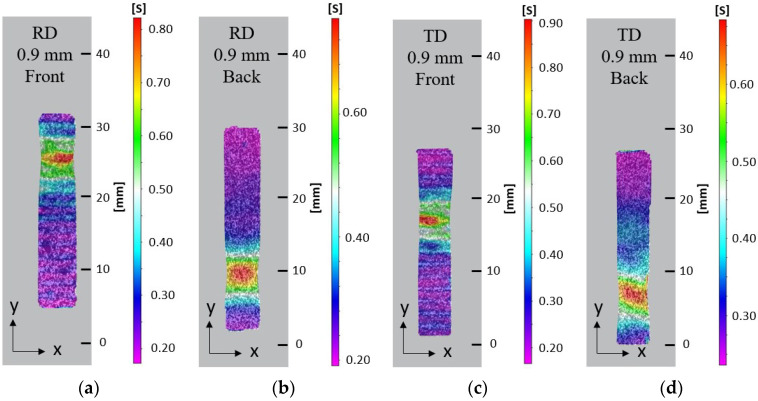
Effective surface strain at 90% extension on the LMD (front) surface and the reverse (back) side with no LMD: (**a**) Front of RD 0.9mm LMD sample; (**b**) Back of RD 0.9 mm LMD sample; (**c**) Front of TD 0.9 mm LMD sample; (**d**) Back of TD 0.9 mm LMD sample.

**Figure 16 materials-15-05985-f016:**
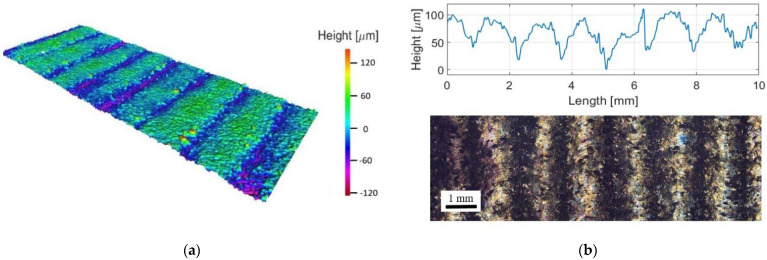
Surface morphology of RD 0.9 mm LMD sample: (**a**) LMD build surface morphology; (**b**) LMD track form with optical image showing unconsolidated powder particles in the grooves between tracks.

**Figure 17 materials-15-05985-f017:**
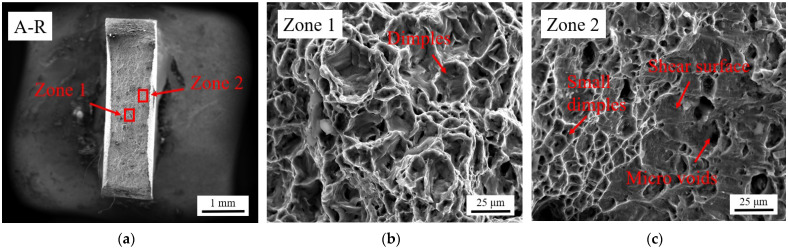
Fractography analysis of A-R material sample: (**a**) Selected areas showing type of fracture at middle (Zone 1) and edge of fracture face (Zone 2); (**b**) High magnification of Zone 1; (**c**) High magnification of Zone 2.

**Figure 18 materials-15-05985-f018:**
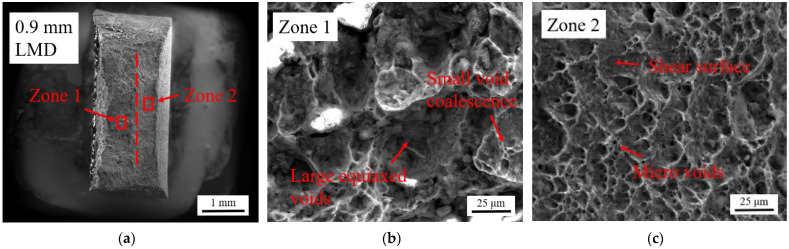
Fractography analysis of 0.9 mm LMD sample: (**a**) Differentiation between LMD added thickness (Zone 1) and non-transformed substrate (Zone 2); (**b**) High magnification of Zone 1; (**c**) High magnification of Zone 2.

**Figure 19 materials-15-05985-f019:**
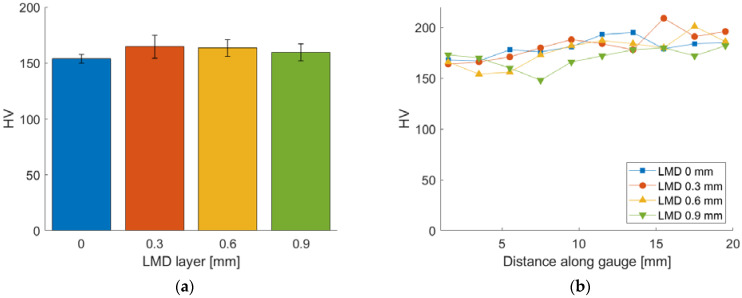
Vickers microhardness results: (**a**) In-plane microhardness of added LMD layer; (**b**) Microhardness along gauge length of fractured tensile specimens.

**Figure 20 materials-15-05985-f020:**
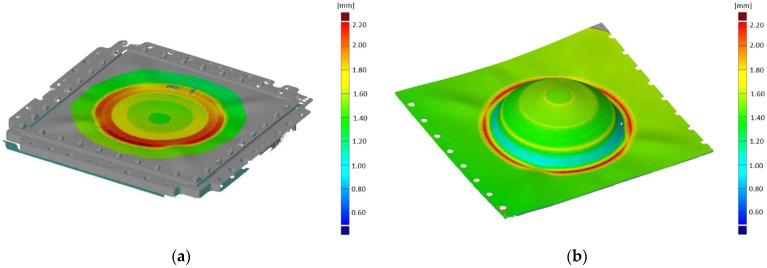
Three-dimensional scans of the LMD tailored SPIF preform and part: (**a**) Thickness distribution across LMD tailored SPIF preform; (**b**) Thickness distribution across LMD tailored SPIF part.

**Figure 21 materials-15-05985-f021:**
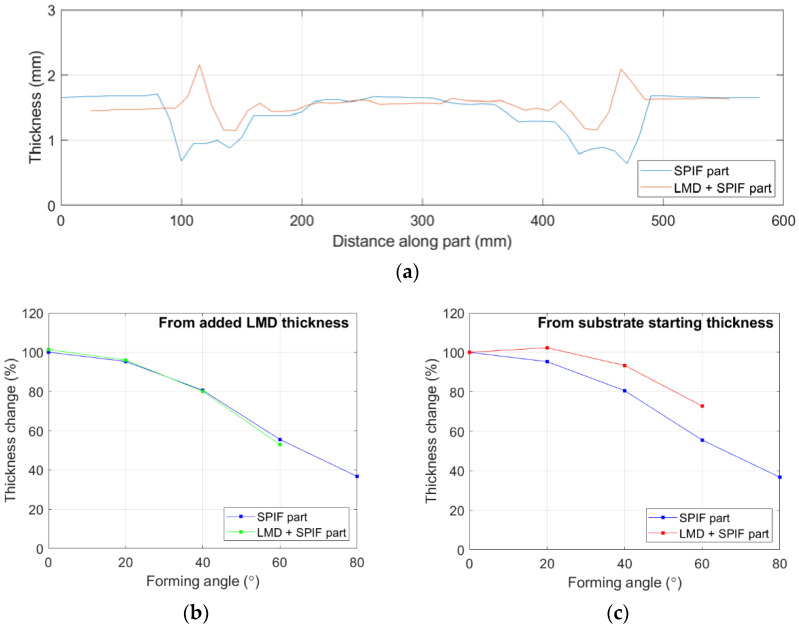
Thickness measurements of SPIF sheets with and without LMD added layer: (**a**) Thickness distribution across SPIF parts; (**b**) % change of each angled section from LMD added thickness; (**c**) % change of each angled section from substrate starting thickness.

**Table 1 materials-15-05985-t001:** Chemical composition of CP-Ti50A.

Constituent Elements	Ti	N	C	H	Fe	O
Composition, wt%	Balance	0.03	0.08	0.015	0.20	0.18

**Table 2 materials-15-05985-t002:** Mechanical properties of A-R CP-Ti50A.

Sheet Orientation	Yield Strength (MPa)	Tensile Strength (MPa)	Poisson’s Ratio
RD	353 ± 1.25	432 ± 0.78	0.33
45° to RD	343 ± 1.62	389 ± 1.21	0.75
TD	378 ± 0.83	420 ± 0.95	0.9

**Table 3 materials-15-05985-t003:** Thickness measurements from the scanned SPIF part to inform the necessary LMD thickening.

Region	Thickness (mm)	Thinning (mm)
θ = 0°	1.65 ± 0.01	N/A
θ = 20°	1.57 ± 0.02	0.07
θ = 40°	1.34 ± 0.03	0.31
θ = 60°	0.93 ± 0.05	0.72
θ = 80°	0.62 ± 0.06	1.03

**Table 4 materials-15-05985-t004:** Material variables.

Material Location	Tensile Gauge Orientation (Y direction)	LMD Thickness (A) (mm)
Set a	RD	0
Set b	RD	0.3
Set c	RD	0.6
Set d	RD	0.9
Set e	45° to RD	0
Set f	45° to RD	0.3
Set g	45° to RD	0.6
Set h	45° to RD	0.9
Set i	TD	0
Set j	TD	0.3
Set k	TD	0.6
Set j	TD	0.9

**Table 5 materials-15-05985-t005:** XRD measurement results.

Material Location	Measurement Direction	Measured Stress (MPa)
Set b	TD	137 ± 6.07
	RD	27 ± 3.97
Set c	TD	93 ± 5.25
	RD	75 ± 5.78
Set d	TD	133 ± 5.45
	RD	55 ± 4.51
Set j	TD	−32 ± 6.21
	RD	0 ± 5.13
Set k	TD	51 ± 5.78
	RD	77 ± 5.47
Set l	TD	168 ± 6.69
	RD	10 ± 4.55

**Table 6 materials-15-05985-t006:** Material properties for CP-Ti50A sheet as-received and with LMD added thickness of variable levels.

Material Location (See Table 4 for Variables)	YS (0.2% Proof Stress) (MPa)	E (% Increase, 20 mm Gauge)	UTS (MPa)
Set a	353 ± 1.25	43 ± 0.75	432 ± 0.78
Set b	362 ± 1.02	38 ± 3.12	393 ± 2.34
Set c	367 ± 0.75	40 ± 3.51	401 ± 0.84
Set d	374 ± 2.36	38 ± 0.93	406 ± 0.39
Set e	343 ± 1.62	52 ± 0.29	389 ± 1.21
Set f	361 ± 0.32	40 ± 3.55	388 ± 1.09
Set g	368 ± 0.98	41 ± 2.85	394 ± 1.41
Set h	377 ± 1.73	38 ± 1.01	407 ± 1.67
Set i	378 ± 0.83	43 ± 0.84	420 ± 0.95
Set j	368 ± 2.49	38 ± 1.16	398 ± 3.63
Set k	373 ± 1.07	37 ± 2.05	402 ± 0.72
Set l	374 ± 1.44	40 ± 3.02	405 ± 1.50

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
