# Peer review of "Tailoring Titanium Sheet Metal Using Laser Metal Deposition to Improve Room Temperature Single-Point Incremental Forming"

_materials, 2022, doi:10.3390/ma15175985_

Round 1

Reviewer 1 Report

Dear authors and editors, 

Please find my comments below: 

Major comments 

1- Please avoid seemingly contradictory phrases. For example, in the abstract authors have mentioned that to achieve Titanium alloy components, an elevated temperature is required. The next phrase introduces potentials for low-temperature forming of Titanium. Therefore, high temperature is not required all the time. 

2- I highly recommend that authors improve the introduction by reporting more relevant works and describing their key findings:

a.     To what material their research had been applied?

b.     Briefly discuss their results in using LMD?

c.     what are the identified gaps? 

3- Introduction, lines 59-60 require reference support. 

4- Please explicitly indicate the novelty of the current manuscript:

a.     Clearly mention what is the novelty

b.     What is the importance of this novelty?

5- Authors have selected two LMD variables. 

a.     It is very important to justify this selection and clearly explain the rationales behind this choice. 

b.     Why did the authors decide to change only these two variables? 

c.     Why not other variables, for example scanning speed? 

d.     Report what did other researchers in similar work choose as variables of interest?

6- What is the added value of Figure 2? It seems like the Figure is not even original and does not provide added value. If it is the case, please consider removing it from the manuscript. 

7- Authors have chosen build heights of 0.3 mm, 0.6 mm, and 0.9 mm. 

a.     What are the rationales for choosing these values? Is it based on calculation? Is it systematic? or are these values just selected to test?

b.     How is it related to the final geometry? what is the limit? 

c.     In another word, how should readers (other researchers) decide about these values for their case?

8- Table 4 contains 12 sets of uniaxial tensile test configurations. Where are those 12 sets exactly located? Are those the ones shown in Figure 4? In this case, Figure 9 only shows 9 regions. Please make sure the reader can easily find where these sets are located in the tests. 

9- Figures 7b and 7c represent three zones (LMD, HAZ, and substrate) after material deposition. 

a.     I appreciate it if the authors can show the original size of the substrate on 7b and 7c if it is deposited material from a portion of the HAZ zone. 

b.     Additional minor comment in this figure, please try to align the bottom of the samples so that they are easier to compare visually. 

c.     I suggest putting the value of the cross-section dimensions in 7a. I recommend reporting the size of the HAZ area (in mm) similar to the LMD layer for 7b and 7c.  

10- Table 5 has 12 rows. Is it related to the 12 sets of configurations mentioned in Table 4? If yes, make sure it is in the same order and consider using the same configuration for more readability. 

11- Please discuss the effect of variables of interests, orientation, (and if possible other fixed variables) on the results in more detail. Could you analyze the interactions of these two variables of interest on the results? 

12- Please summarize the weaknesses and limitations of your studies in the conclusion section.

13- It is very important that the authors explain the perspective and possible future directions of this research. What can be done next by authors or by other researchers? 

Minor comments: 

14- Fonts in tables are not uniform (for example, see tables 2, 3, 4, and 5).

15- Please avoid putting the unit in equation 1. Simply remove the unit (J/mm). 

16- There is an error in the text (Line 174).

17- In line 293 a dash is missing between 2.4% and 5.6% and in line 11 for low-temperature forming

Author Response

Comment 1:

Please avoid seemingly contradictory phrases. For example, in the abstract authors have mentioned that to achieve Titanium alloy components, an elevated temperature is required. The next phrase introduces potentials for low-temperature forming of Titanium. Therefore, high temperature is not required all the time. 

Response to comment 1:

Re-written

“Typically, due to their limited formability, elevated temperatures are required in order to achieve complex shapes in Titanium alloys. However, there are opportunities for forming such alloys at room temperature using incremental forming processes such as single-point incremental forming (SPIF). “

Comment 2:

I highly recommend that authors improve the introduction by reporting more relevant works and describing their key findings:

  1. To what material their research had been applied?
  2. Briefly discuss their results in using LMD?
  3. what are the identified gaps? 

Response to comment 2:

Changes made. The materials used in the referenced works have been added which highlights the lack of research into SPIF of titanium as most studies use aluminium sheet for their investigations. The need to improve low temp forming of titanium and how SPIF is a cost-effective solution to this problem is made clear. References added to clarify advantages brought by LMD. Knowledge gaps made clearer in introduction. 1st - SPIF of pure titanium at room temperature not sufficiently studied. Typically, materials include aluminium and steel. References to this lack of material variety has been added to introduction. 2nd - Use of LMD to reinforce SPIF preform with variable thickness a novelty. Single thickness attempted in another study (referenced). However, attempt to account for different thinning across multiple forming angles is novel. Made clear in introduction.

Comment 3:

Introduction, lines 59-60 require reference support.

Response to comment 3:

Re-wrote small section to ensure references where highlighted and clear. Further references added to expand on literature study.

Comment 4:

Please explicitly indicate the novelty of the current manuscript:

  1. Clearly mention what is the novelty
  2. What is the importance of this novelty?

Response to comment 4:

Added section to explain novelty final paragraph of introduction, see below.

“The purpose of this investigation was to determine if locally thickening a pure titanium sheet using LMD could be used to compensate the issue of wall thinning in SPIF. The development of this novel hybrid process is important as it may potentially provide a route to improve geometric accuracy in cold formed titanium sheet parts.”

Comment 5:

Authors have selected two LMD variables. 

  1. It is very important to justify this selection and clearly explain the rationales behind this choice. 
  2. Why did the authors decide to change only these two variables? 
  3. Why not other variables, for example scanning speed? 
  4. Report what did other researchers in similar work choose as variables of interest?

Response to comment 5:

  1. Two variables. Justification added to introduction.
  • Build height. Extent of thinning in respect to forming angle is key to investigation.
  • LMD build orientation. Anisotropy of a-r sheet is to be studied, and how LMD alters this.
  1. b) Comparison with previous SPIF part used for validation. Shape of part kept consistent to allow for this.
  2. c) LMD variables were held constant because we did not want any confounding factors to influence the results. Previous works had identified the scan speeds etc used in the LMD as optimum for this purpose.
  3. d) Shape of the part adjusted in one study. Toolpath optimisation investigated in another study. Other variables tested include tool size, step down, scanning speed. More information regarding these studied variables has been added to introduction.

Comment 6: 

What is the added value of Figure 2? It seems like the Figure is not even original and does not provide added value. If it is the case, please consider removing it from the manuscript. 

Response to comment 6:

Figure 2 removed from manuscript.

Comment 7:

Authors have chosen build heights of 0.3 mm, 0.6 mm, and 0.9 mm. 

  1. What are the rationales for choosing these values? Is it based on calculation? Is it systematic? or are these values just selected to test?
  2. How is it related to the final geometry? what is the limit? 
  3. In another word, how should readers (other researchers) decide about these values for their case?

Response to comment 7:

  1. A GOM scan of a previously formed SPIF part was made and the thinning quantity was used to define what additional thickness was required. This has been clarified in the revision.
  2. The scan justifies what thickness of reinforcement is necessary to thicken to achieve homogeneous thickness in final part, something not possible when performing SPIF with a typical flat sheet preform.
  3. Predictions on level of thinning, potentially from previous experimental SPIF trials/ SPIF simulations/ or literature, should be made to inform level of thickness reinforcement is required.

Comment 8:

Table 4 contains 12 sets of uniaxial tensile test configurations. Where are those 12 sets exactly located? Are those the ones shown in Figure 4? In this case, Figure 9 only shows 9 regions. Please make sure the reader can easily find where these sets are located in the tests. 

Response to comment 8:

A new figure (3) was made to organise all material samples and reference material variables. All references to material variable testing configurations, tables 4, 5 and 6, now refer directly to figure 3. Makes it easier for reader to find where all sets located.

Comment 9:

Figures 7b and 7c represent three zones (LMD, HAZ, and substrate) after material deposition. 

  1. I appreciate it if the authors can show the original size of the substrate on 7b and 7c if it is deposited material from a portion of the HAZ zone. 
  2. Additional minor comment in this figure, please try to align the bottom of the samples so that they are easier to compare visually. 
  3. I suggest putting the value of the cross-section dimensions in 7a. I recommend reporting the size of the HAZ area (in mm) similar to the LMD layer for 7b and 7c.  

Response to comment 9:

Figure has been adjusted to show dimensions and bottom surfaces aligned as suggested.

Comment 10:

Table 5 has 12 rows. Is it related to the 12 sets of configurations mentioned in Table 4? If yes, make sure it is in the same order and consider using the same configuration for more readability.

Response to comment 10:
Tables have been made more consistent and all link back to original diagram, figure 3 for better readability. See comment 8.

Comment 11:

Please discuss the effect of variables of interests, orientation, (and if possible other fixed variables) on the results in more detail. Could you analyze the interactions of these two variables of interest on the results? 

Response to comment 11:

Tensile results indicate when LMD tailored material subjected to uniaxial deformation the LMD tailoring changes its properties, however the orientation and build height variations have limited effect (perform very similarly). The isotropic behaviour of LMD tailored material and lack of influence of build height on properties mentioned in results section. As such, anisotropy of LMD preform for performing SPIF with tailored sheet of less relevance. This influenced the LMD tailored preform design and subsequent SPIF trial. This has been added to conclusions.

Comment 12:

Please summarize the weaknesses and limitations of your studies in the conclusion section.

Response to comment 12:

  • Reduced ductility and lower formability of LMD tailored material is now stated in concluding bullet points.
  • SPIF LMD hybrid part early failure is discussed in detail in results section.
  • Planned subsequent quality assessment study is required and so now mentioned in closing statement regarding future work.

Comment 13:

It is very important that the authors explain the perspective and possible future directions of this research. What can be done next by authors or by other researchers? 

Response to comment 13:

  1. “Further analysis of the LMD + SPIF part is planned to help understand the cause of this early fracture and determine the potential in this hybrid LMD SPIF process.”
  2. “A subsequent quality assessment of the final sheet part is planned.”

Above two future directions of work by authors have been provided at end of results and conclusions sections.

Comment 14.

Fonts in tables are not uniform (for example, see tables 2, 3, 4, and 5).

Response to comment 14:

Fonts fixed

Comment 15:

Please avoid putting the unit in equation 1. Simply remove the unit (J/mm). 

Response to comment 15:

Equation removed, deemed unnecessary.

Comment 16:

There is an error in the text (Line 174).

Response to comment 16:

Section adjusted and error fixed.

Comment 17:

In line 293 a dash is missing between 2.4% and 5.6% and in line 11 for low-temperature forming

Response to comment 17:

Errors fixed.

Reviewer 2 Report

1. This paper aims to improve room temperature single-point Incremental forming by laser metal deposition prior to SPIF. But the authors only characterized the microstructural and mechanical properties of a CP-Ti50A sheet tailored with local thickening via CP titanium (Grade 2) powder deposited using laser metal deposition. And the SPIF of the tailored CP titanium sheet will be planned in the future. Regarding to the laser metal deposition of titanium and its alloys, it has been widely studied. Therefore, what is the novelty of this paper? Is it appropriate to show “improve room temperature Single-Point Incremental Forming” in the title, since the feasibility of this process is still not validated? What’s more, the LMD material exhibited reduced ductility and lower formability to the A-R material

2. " Typically, ductile fracture mechanism is preceded by substantial plastic deformation and includes the growth and coalescence of microscopic voids which may nucleate from hard inclusions which do not deform at the same rate as the bulk material. " In this sentence, what are the hard inclusions for CP titanium (Grade 2) ?

Author Response

Comment 1:

This paper aims to improve room temperature single-point Incremental forming by laser metal deposition prior to SPIF. But the authors only characterized the microstructural and mechanical properties of a CP-Ti50A sheet tailored with local thickening via CP titanium (Grade 2) powder deposited using laser metal deposition. And the SPIF of the tailored CP titanium sheet will be planned in the future. Regarding to the laser metal deposition of titanium and its alloys, it has been widely studied. Therefore, what is the novelty of this paper? Is it appropriate to show “improve room temperature Single-Point Incremental Forming” in the title, since the feasibility of this process is still not validated? What’s more, the LMD material exhibited reduced ductility and lower formability to the A-R material

Response to comment 1:

It has been decided to include the SPIF trial into this manuscript to justify the title and novelty of introducing LMD. See highlighted sections in adapted manuscript. This justifies the title of the paper and provides the desired novelty to the work.

Comment 2:

“Typically, ductile fracture mechanism is preceded by substantial plastic deformation and includes the growth and coalescence of microscopic voids which may nucleate from hard inclusions which do not deform at the same rate as the bulk material. " In this sentence, what are the hard inclusions for CP titanium (Grade 2)?

Response to comment 2:

This is not stating there are hard inclusions within the CP-Ti50A material, just explaining mechanism. However, to avoid confusion it has been removed.

Reviewer 3 Report

Dear Authors,

The work presented in the paper entitled “Tailoring Titanium Sheet Metal using Laser Metal Deposition to Improve Room Temperature Single-Point Incremental Forming” is excellent and novel.

However, before final publication, some points need to be addressed. The details of the comments are as follows.

1) Line 32 – “Single point incremental forming (SPIF) is a method of producing sheet metal parts from materials that are often difficult to form at room temperature without the need for expensive tooling”, Provide appropriate reference.

2) Line 53 – “There have been studies into tailoring sheet blanks to optimise their functionality”. What study?

3)     

Line 54 – “One such study designed sheet”. Rewrite the sentence

4) Line 88 – Why did the authors choose pure titanium grade 2 (CP-Ti50A) with 1.6mm thickness?

5) Provide mechanical properties of the materials mentioned above?

6) What about the density of titanium grade 2 (CP-Ti50A)? Is there any effect during the LMD process?

7) Provide X-ray diffraction (XRD) results?

8) What standards are followed for the tensile specimen (Figure 6)?

9) Line 204 – “It was 204 impossible to measure the temperature during deposition”. Please see the below-mentioned link

https://www.optris.global/3d-printing-additive-manufacturing

I think it is possible to measure the temperature.

10) Line 212 – “This is likely an effect of residual heat and the sample being exposed to a higher temperature for longer”. Only residual heat effect or any other effect? Please check again.

Author Response

Comment 1:

Line 32 – “Single point incremental forming (SPIF) is a method of producing sheet metal parts from materials that are often difficult to form at room temperature without the need for expensive tooling”, Provide appropriate reference.

Response to comment 1:

Reference added.

Comment 2:

Line 53 – “There have been studies into tailoring sheet blanks to optimise their functionality”. What study?

Response to comment 2:

Two studies were referenced, (Raut, 2020) and (Bambach, 2017). Sentence adjusted to make clearer.

Comment 3:

Line 54 – “One such study designed sheet”. Rewrite the sentence

Response to comment 3:

Re-wrote sentence.

Comment 4:

Line 88 – Why did the authors choose pure titanium grade 2 (CP-Ti50A) with 1.6mm thickness?

Response to comment 4:

1.6 mm thickness allows previous SPIF part to be used for comparison / validation.

Comment 5:

Provide mechanical properties of the materials mentioned above?

Response to comment 5:

Table containing yield strength, ultimate tensile stress and Poisson’s ratio added for CP-Ti50A.

Comment 6:

6) What about the density of titanium grade 2 (CP-Ti50A)? Is there any effect during the LMD process?

Response to comment 6:

Although density measurements were not carried out, microstructural investigations confirmed absence of the pores and it is assumed that the density of the LMD materials is very close to as received.

Comment 7:

Provide X-ray diffraction (XRD) results?

Response to comment 7:

XRD results added.

Comment 8:

What standards are followed for the tensile specimen (Figure 6)?

Response to comment 8:

Design based on ASTM-E8/EM. Adapted for tooling at AFRC. Comment added to clarify this.

Comment 9:

Line 204 – “It was 204 impossible to measure the temperature during deposition”. Please see the below-mentioned link

Response to comment 9:

With the equipment at Optris yes it would have been possible. However, the setup at the external suppliers who performed the LMD could only facilitate the AM process, the available equipment did not allow for temperature measurements to be made.

Comment 10:

Line 212 – “This is likely an effect of residual heat and the sample being exposed to a higher temperature for longer”. Only residual heat effect or any other effect? Please check again.

Response to comment 10:

Long exposure time of laser during LMD. Sentence changed to reflect this.

Round 2

Reviewer 2 Report

Authors have addressed my comments properly.The paper is acceptable.

Reviewer 3 Report

The authors have been incopotated the comments raised by the reviewer. I hope paper will accept for publications.